# miRNAs as Biomolecular Markers for Food Safety, Quality, and Traceability in Poultry Meat—A Preliminary Study

**DOI:** 10.3390/molecules29040748

**Published:** 2024-02-06

**Authors:** Nada Baraldo, Luna Buzzoni, Luisa Pasti, Alberto Cavazzini, Nicola Marchetti, Annalaura Mancia

**Affiliations:** 1Department of Chemical, Pharmaceutical and Agricultural Sciences (DoCPAS), University of Ferrara, 44121 Ferrara, Italy; nada.baraldo@unife.it (N.B.); alberto.cavazzini@unife.it (A.C.); 2Department of Life Science and Biotechnologies, University of Ferrara, via L. Borsari 46, 44121 Ferrara, Italy; luna.buzzoni@unife.it; 3Department of Environmental and Prevention Sciences, University of Ferrara, via L. Borsari 46, 44121 Ferrara, Italy; luisa.pasti@unife.it; 4Council for Agricultural Research and Economics, via della Navicella 2/4, 00184 Rome, Italy; 5Department of Biology and Marine Science, Marine Science Research Institute, 2800 University Blvd N, Jacksonville, FL 32211, USA

**Keywords:** non-coding RNA, chicken breast, animal health, miR-21, miR-126, biomarkers

## Abstract

In this study, the expression and abundance of two candidate chicken (Gallus gallus; gga) microRNAs (miRNAs, miR), gga-miR-21-5p (miR-21) and gga-miR-126-5p (miR-126), have been analyzed in order to identify biomarkers for the traceability and quality of poultry meat. Two breeds of broiler chickens were tested: the most common Ross308 (fast-growing) and the high-quality Ranger Gold (slow-growing). A preliminary analysis of the two miRNAs expressions was conducted across various tissues (liver, lung, spleen, skeletal muscle, and kidney), and the three tissues (lung, spleen, and muscle) with a higher expression were chosen for further analysis. Using quantitative reverse transcription polymerase chain reaction (RT-qPCR), the expression of miRNAs in the three tissues of a total of thirteen animals was determined. The results indicate that miR-126 could be a promising biomarker for the lung tissue in the Ranger Gold (RG) breed (*p* < 0.01), thus suggesting a potential applicability for tracing hybrids. RG exhibits a significantly higher miR-126 expression in the lung tissue compared to the Ross308 broilers (R308), an indication of greater respiratory capacity and, consequently, a higher oxidative metabolism of the fast-growing hybrid. During sampling, two R308 broilers presented some anomalies, including airsacculitis, hepatic steatosis, and enlarged spleen. The expression of miR-126 and miR-21 was compared in healthy animals and in those presenting anomalies. Chickens with airsacculitis and hepatic steatosis showed an up-regulation of miR-21 and miR-126 in the most commercially valuable tissue, the skeletal muscle or breast (*p* < 0.05).

## 1. Introduction

In recent decades, there has been a significant increase in the demand for poultry meat due to low production costs, highly efficient feed conversion, and the technological, nutritional, and sensory characteristics of the finished products [1]. Furthermore, current forecasts and projection studies anticipate that the expansion of the poultry market will continue to grow in the future. Currently, poultry meat represents the second highest consumed meat globally [2,3]. This growing demand has led to progressive improvements in genetic selection to produce fast-growing broiler chickens, resulting in the emergence of numerous spontaneous idiopathic muscular anomalies along with increased susceptibility to stress-induced myopathy [1,4]. In view of these development issues with swift-raised chickens, slow-growing hybrids have been obtained and selected with regard to meat production with improved texture and flavor characteristics, also meeting animal welfare guidelines and with constant attention to the animals’ needs, as proposed by the Welfare Quality protocol [5,6]. Poultry meat is, today, mainly obtained from two broiler breeds: the fast-growing (i.e., R308) and the slow-growing (i.e., RG). RG grows at a rate of approximately 50 g per day, taking about 25 days longer to mature compared to traditional chickens. Standard chickens in commercial production systems, like the R308, undergo rigorous genetic selection to achieve a rapid growth rate and to develop a smaller muscle mass than RG. They reach slaughter weight in 30–40 days, as opposed to the minimum of 81 days for RG [7]. Given its slower growth rate, this broiler (RG) complies with animal welfare labeling. Several authors have concluded that slow-growing broiler strains perform better in terms of health issues compared to conventional strains [8]. The growth rate, age, and live body weight influence gait and the development of lameness. In fact, in slow-growing broilers, almost no gait alterations have been observed, along with a lower incidence of tibial dyschondroplasia and lower mortality rates compared to fast-growing broilers [9,10,11,12].

Both quality and traceability of poultry meat, as well as genetic characteristics of broilers, have been studied using omics technologies, particularly by measuring the expression of miRNAs in different tissues. It has been already established that miRNAs act as coordinator biomolecules in cellular processes and gene expression, including the modulation of animal development, homeostasis, immune responses, infection control, and are also crucial for regulating stem cell self-renewal and tissue differentiation [6,13,14,15]. In addition, the tissue-specific expression of miRNAs makes them promising biomarkers for monitoring the animal wellbeing and its genetic traits that may predispose it to diseases and myopathies. Preserving the clinical health of farm animals, avoiding diseases and environmental stress, plays a crucial role in producing safe and high-quality food [6].

Recent studies investigated the regulation of immunity through miRNAs, as influenced by the interaction between the host and the pathogen [16]. In addition, it has been found that miRNAs can be influenced by diet, such as selenium deficiency (e.g., as revealed by miR-33-3p), that can promote tissue damage, including cellular apoptosis, through the triggering of oxidative stress and miRNA expression [17]. In this context, miRNAs play the role of ideal biomarkers, providing detailed information on cellular mechanisms and animal adaptation to adverse events. This offers insights into the health status of farmed animals and, consequently, it defines a new perspective of the meat quality.

For the sake of completeness, miRNAs are gaining more and more significance not only in the case of quality, safety and traceability for meat products. Several original research studies reviewed by Rani et al. [18] have focused on the identification of miRNAs in milk, because they constitute bioactive components associated with natural nanovesicles known as exosomes. The miRNAs encapsulated in exosomes serve as mediators of communication between cells, influencing the physiology of target cells and being less prone to degradation. Consequently, milk encapsulated miRNAs remain stable during the digestive processes, they can permeate the intestinal barrier, enter tissue cells and exert their bioactive function. In particular, in a study conducted by Chen et al. [19], they collectively identified about 245 miRNAs in raw milk, with an essential role in evaluating the milk quality. These miRNAs are characterized by their specific expression and low susceptibility to manipulation, making them reliable indicators in the event of potential adulteration. This finding holds great importance, considering that traditional biomarkers for milk quality, such as proteins, fats, or carbohydrates, exhibit a limited effectiveness in providing precise indications of the product’s quality [18,19]. Often, identification of frauds and adulteration in foods require comprehensive chemical characterization of the nutritional components. The use of miRNAs as quality biomarkers proves to be especially meaningful because of their capacity to offer a more accurate assessment compared to conventional markers. Another valuable key point is that these miRNAs are specific for bovine milk, facilitating the identification of adulterated or low-quality milk.

On the contrary, the number of literature studies addressing the issue of chicken meat traceability using miRNAs as biomarkers is still limited; although, this is an increasingly important topic for consumers, authorities and food industries. Traceability is essential for food authentication, for ensuring its safety and detecting possible frauds [20,21]. Currently, there is no standard or universal method to identify frauds; therefore, the primary goal is to identify specific biomarkers capable of revealing and validating potential food adulterations. At the same time, it is crucial to develop methods of analysis for biomarkers recognized by regulatory agencies as unique, reproducible, and effective target biomolecules to be determined [22,23]. miRNAs have the potential to become fundamental biomarkers that would allow food testing for the presence of undeclared species in meat-based products, ensuring highly reliable results.

This study is hence aimed at identifying miRNA biomarkers for the traceability of two broiler breeds (i.e., R308 and RG). Here, two miRNAs of interest, miR-126 and miR-21, were selected, and their expression levels were assessed using RT-qPCR in three distinct tissues: spleen, lung, and skeletal muscle (breast). The expression of both miRNAs is extensively documented in the scientific literature. In chickens, miR-21-5 exhibits high expression levels in broiler muscular tissue both during the embryonic period and after birth. Furthermore, in humans, previous studies have highlighted the role of miR-21 in diseases such as cardiovascular pathologies and cancer [24,25]. Additional research indicates that miR-21 plays a significant role in inducing the migration of lung smooth muscle cells under hypoxic conditions [26]. Further studies have linked miR-21 to the regulation of vascular and cardiac smooth muscle, as well as skeletal muscle formation and regulation, though further investigation is needed to deepen this correlation [25,27]. On the other hand, miR-126 is known for its predominant expression in highly vascularized tissues such as the lung, heart, liver, hematopoietic cells, and endothelial cells [28,29,30,31,32,33]. In the lung, miR-126 shows significantly higher expression levels compared to other tissues, suggesting a substantial role within this organ. Some studies indicate miR-126’s under-regulation in lung carcinomas, suggesting its crucial involvement in maintaining the lung’s normal state [33,34].

The purpose of this study was not only to identify useful biomarkers for poultry traceability, food safety, and quality. Furthermore, it was hypothesized that miR-21 and miR-126 would be expressed differently between tissues of two broilers, RG and R308 and within tissues of Ross308 broilers affected by pathologies.

## 2. Results

### 2.1. Validation of the Reference Gene to Measure miRNA Expression

In total, two housekeeping genes (5S rRNA and U6 snRNA) were selected from the literature; these genes are typically used as reference genes for accurate miRNA normalization. Comparing the transcription levels of the housekeeping genes, the CT values were directly plotted, assuming a similar threshold for all of the genes evaluated. The U6 snRNA housekeeping gene exhibited a greater stability compared to the 5S rRNA gene; therefore, the normalization of data involved averaging the two housekeeping genes (Appendix A). The CT is defined as the number of cycles required for the fluorescence to reach a specific threshold level of detection and is inversely correlated with the quantity of miRNA present in the reaction. The CT values for all of these genes showed a normal distribution, according to the Kolmogorov–Smirnov method in each analyzed tissue [35,36].

### 2.2. Analysis and Quantification of the Presence of Two Specific miRNAs

In the first pilot experiment, miRNAs were extracted in the muscle (breast), liver, kidney, lung, and spleen, demonstrating excellent efficiency (quality/quantity ratio of total miRNA) across all tissues in both breeds. The expression of both examined miRNAs was detected in all tissues with a similar expression in the muscle tissues of both breeds and was slightly more pronounced in the muscle of RG than R308 (Figure 1). Remarkably, in both breeds, miRNA-21 was strongly expressed in spleen (Figure 1A), while miRNA-126 was strongly expressed in the lung (Figure 1B), with higher values in RG compared to R308. For this reason, in the second sampling procedure, the lung and the spleen tissues were collected and further analyzed together with the muscle tissue.

### 2.3. Determination of Tissue Specificity for miRNA Biomarkers

After an initial screening of miRNA expression in all tissues of the two breeds, the spleen, lung, and muscle were selected for further analysis on a larger set of samples. The spleen was selected for the high expression of miR-21 in both breeds; the lung was selected for the high expression of miR-126 in both breeds; and the muscle was selected for its commercial value, and the relative expression of both miRNAs in both breeds. The expression levels reported in terms of fold change (FC) for each studied miRNA in each bread are reported in Appendix A and shown in Figure 2.

Quantification of miR-21 and miR-126 in the tissues of the two breeds shows significant difference in the expression of miR-126 in the lung tissue between the two breeds linked to the higher expression in the RG chickens (*p* < 0.01). We hypothesize that miR-126 could be a pulmonary biomarker for tracing the RG hybrid compared to the R308 broiler (Figure 2).

The literature studies have suggested that miR-126 might play a particularly significant role in the lung, as it is typically highly expressed in lung tissues [24]. Our findings of a markedly higher expression in RG lungs compared to R308 broilers suggest that the fast-growing hybrid could have greater respiratory capacity and consequently possesses a higher oxidative metabolism.

### 2.4. Expression of miRNAs in Healthy and Diseased R308 Broilers

Right after slaughter, during sampling, the animals were examined, and their health status was overall determined. A total of four R308 chickens exhibited anomalies such as airsacculitis, hepatic steatosis, and an enlarged spleen. However, among these, two R308 chickens, despite having airsacculitis and an enlarged spleen, did not show significant differences in relative expression compared to the healthy R308 chickens, and were thus considered healthy. On the other hand, the remaining two chickens, which presented airsacculitis (along with diarrhea) and hepatic steatosis, displayed a much higher relative expression, and therefore, they were classified as diseased. Subsequently, the animals were divided into two groups (healthy and diseased), and the relative expression of miR-21 and miR-126 was compared in the tissues (spleen, muscle, and lung) of the two groups.

The expression of both miRNAs appeared significantly upregulated in the diseased group in the muscle (breast) tissue (*p* < 0.05), while the spleen and lung tissues did not reveal statistically relevant differences in the miRNA expression. These outcomes can suggest a correlation between up-regulation of miR-21 and miR-126 and evidence of airsacculitis and hepatic steatosis in broilers (Figure 3).

## 3. Discussion

Presently, it is part of the standard laboratory procedures that the relative quantification of miRNA expression needs to be compared with housekeeping genes (i.e., two in this work, see Section 4.4) expected to exhibit stable expression within the same sample [37,38]. The high sensitivity of RT-qPCR requires proper normalization to correct bias from non-biological variations, and in this sense, the use of housekeeping genes turns into the most commonly employed method [39,40]. However, there are no universal housekeeping genes for normalizing miRNAs through RT-qPCR analysis. Considering the importance of obtaining accurate data to identify reliable miRNA biomarkers for food traceability, the mean values of Ct (quantification cycle) for the housekeeping genes (5S rRNA and U6 snRNA) were calculated. The Ct values showed a normal distribution in all three of the analyzed tissues. This allowed for the normalization of amplification data across multiple endogenous controls to ensure miRNA quantification without being influenced by experimental variables [41].

From the statistical analysis of the three tissues, it emerged that miR-126 could be a valid biomarker for Ranger Gold traceability, as it exhibits significant expression in the lung (*p* < 0.01) compared to the broiler Ross-308. To our knowledge the miRNA analyzed in this study do not show sex-specific expression in the tissues tested [42,43,44]. Previous studies indicate that miR-126 is significantly expressed at a higher ratio compared to other tissues [33] and is highly conserved in vertebrates with a high expression in the vascular system such as blood islands (in the embryo representing the primary site of vascular development and hematopoiesis), blood vessels, and the heart tube [45,46,47]. Therefore, it appears to be a pro-angiogenic miRNA, consistent with the results due to its high tissue-specific expression in the highly vascularized lung organ. A recent study indicated that miR-126 acts on the innate immunity of chickens. In fact, researchers discovered that miR-126 significantly inhibited the expression of inflammatory genes related to cytokines after a Newcastle disease virus (NDV) infection. miR-126 notably promoted NDV replication, suggesting that it regulates the host’s immune response by suppressing the ‘cytokine storm’ after viral infection. This negative regulation effectively prevents damage to the host caused by an excessive immune response [48]. Ishizaki et al. have demonstrated that miR-126 regulates ERK signaling and cytokine production by directly targeting Spred1 (Sprouty-related Ena/VASP homology-1 domain-containing protein). The latter negatively regulates FcεRI-mediated cytokine production in mast cells [49].

In this study, it was shown that fast-growing chickens with evident symptoms, such as airsacculitis or fatty liver disease, exhibit a significant difference in the expression of miR-126 in skeletal muscle compared to healthy broilers (*p* < 0.05). This could serve as an indicator or marker of animal illness. The literature reports its direct role in regulating skeletal muscle growth and insulin-like growth factor 1 (IGF-1) activation, identified in selected breeds of fast-growing meat chickens [50,51]. Moreover, the role of miR-126 in vascular integrity and angiogenesis is well established, acting on enhancing the action of VEGF and FGF, which serve as potent inducers of angiogenesis. Meanwhile, Spred-1, an intracellular inhibitor of the Ras/MAP kinase pathway, serves as a target for repression [32]. The hypothesis inferred from the overexpression of miR-126 is that there is stimulation of the angiogenesis process due to tissue damage and/or a microorganism.

Similarly, miR-21 was found to be overexpressed in fast-growing chickens that exhibited airsacculitis and hepatic steatosis. In a study, miR-21 has been recently identified as an upregulated miRNA in chicken-interdigital regression associated with inflammation, cellular senescence, and programmed cell death [52]. It has been found that miR-21 is highly expressed in many species and is associated with heart diseases and a wide variety of human tumors [53,54]. In another study, miR-21 was found to be upregulated in low-weight chickens [55]. Different levels of the miR-21 expression have been identified in the skeletal muscle of chickens at various growth stages [56] and in the skeletal muscle of broiler chickens and laying hens [57]. A recent study demonstrated that miR-21 can inhibit cell proliferation [58], suggesting that it might act as a negative regulatory factor for chicken growth. These highly differentially expressed miRNAs could play a key role in chicken growth traits and might be used as candidate genes for further studies [59]. Based on these latter studies found in the literature, it can be hypothesized that the overexpression of miR-21 might be an indicator of cell death and inflammation. Future studies will aim to investigate the tissue damage caused by the likely presence of a bacterium or virus, which also compromises the quality of the food product.

## 4. Materials and Methods

### 4.1. Animals and Tissues

A total of seven fast-growing R308 broilers and six slow-growing Ranger Gold broilers originating from a local slaughterhouse (Avicola Artigiana Srl di Bersanetti Mauro & C. S.n.c., Formignana, FE, Italy) were used in this study (Table 1). Randomly selected individuals from R308 and RG were sacrificed using electrical water bath stunning followed by bleeding on day 48 for R308 and day 71 for RG post-hatching. The birds were caught and kept in transport boxes for 2–3 h before stunning. Muscle, liver, kidney, lung, and spleen were collected from one animal of each breed in a first sampling round. Lung, spleen, and muscle were collected from 6 R308 (4 males, 2 female) and 5 RG (5 females) and stored in 2 mL of stabilization buffer (RNAlater), then stored at −20 °C until further processing. All methods were carried out following relevant guidelines and regulations (Regulation (EC) No. 1099/2009).

### 4.2. miRNA Extractions

For the extraction of miRNAs from chicken tissues the phenol-guanidinium isothiocyanate method was used. The samples were homogenized in TRI-Reagent (Invitrogen, Waltham, MA, USA; 1 mL of TRI Reagent solution per 100 mg of tissue) and incubated at room temperature for 5 minutes (min). The presence of guanidinium isothiocyanate ensures the inhibition of any present RNases, thus preserving the RNA from degradation during extraction. Chloroform (200 μL per mL of TRI-Reagent) was added to the homogenized sample, and through centrifugation at 12,000× *g* for 15 min at 4 °C, the sample was separated into three phases: an aqueous phase containing RNA to be collected (400 μL); an interface containing DNA; and an organic phase containing proteins. Subsequently, ethanol (215 μL per mL of TRI-Reagent) was added to the collected aqueous phase. Total RNA was then separated from small RNAs using the PureLink miRNA Isolation kit (Thermo Fisher Scientific, Waltham, MA, USA) according to manufacturer’s instructions. Briefly, each sample was transferred into a spin cartridge and centrifuged for 1 min at 12,000× *g*. Next, 70% ethanol was added to allow small RNA molecules to bind to the silica column of the cartridge. Impurities were removed through washing with a wash buffer. Small RNA (including miRNAs and siRNAs) was eluted in 50 μL of RNase-free sterile water and stored at −80 °C.

### 4.3. Quality Control small RNA

An aliquot of RNA was used for sample quality control using a spectrophotometer (BioSpec-nano Spectrophotometer) with the measurement of the sample absorbance at 260 nm (for nucleic acids), 280 nm (for proteins), and 230 nm (for organic compounds such as phenols or extraction residues). By evaluating the ratio of optical density values obtained from the different absorbances, it is possible to determine if the sample is contaminated with proteins or organic solvents that could interfere with reverse transcription reactions for gene expression or miRNAs experiments. For an uncontaminated RNA sample, the A260/280 ratio should fall between 1.8 and 2.1, while the A260/230 ratio should exceed 2 (Table 1).

### 4.4. Reverse Transcription Quantitative Real-Time PCR (RT-qPCR)

For reverse transcription (RT), the miRCURY LNA universal RT kit (Qiagen, Germantown, MD, USA) was used, following the manufacturer’s protocol. A Bio-Rad T100 Thermal Cycler was employed for this process. RT was carried out as per the manufacturer’s protocol. The RNA was adjusted to a concentration of 5 ng/μL. The reaction volume was 10 μL, comprising 2 μL of RNA template (5 ng/μL), 2 μL of reaction buffer, 1 μL of enzyme mix (10×), 4.5 μL of RNase-free water, and 0.5 μL of UniSp6 RNA spike-in (Qiagen, Germantown, MD, USA), which is a synthetic exogenous-control RNA.

Quantitative RT-PCR (qRT-PCR) assays for miRNA expression analysis utilized the manufacturer’s protocol with optimizations made for the cDNA concentration. The assays were performed using SYBR Green chemistry in a Bio-Rad CFX Connect Real-Time System thermal cycler, employing 2× miRCURY SYBR Green master mix and hsa-miR-126-5p or hsa-miR-21-5p primers (miRCURY LNA™ miRNA PCR Assay, targeting respectively mature miRNA gga-miR-126-5p and gga-miR-21-5p), according to the manufacturer’s instructions. The qRT-PCR test was run in triplicates. In a 96-well PCR plate (Bio-Rad, Hercules, CA, USA), 6 μL of reaction mixture, composed of 5 μL of SYBR Green real-time PCR master mix and 1 μL of miRNA LNA™ primer mix, was dispensed first. This was followed by the addition of 3 μL of diluted cDNA (1:10 dilution). The reaction conditions for miRNA amplification were as follows: 95 °C for 2 min followed by 40 cycles of 95 °C for 10 seconds (secs) and 56 °C for 60 secs. The miRNA expression level was measured using the quantification cycle (Cq) value. A total of two housekeeping genes, 5S rRNA and U6 snRNA (miRCURY LNA™ miRNA PCR Assay), were selected as internal controls. The relative expression of mRNA was estimated using the 2−∆∆Ct method [60].

### 4.5. Statistical Analysis

The data were collected and analyzed using the Bio-Rad CFX Manager™ software ver. 3.1 (rev. 1517.0823). The relative expression levels of miRNAs were then normalized using the cycle threshold (Ct) for each miRNA and the 2−∆∆Ct method. Briefly, the ∆Ct value was calculated by subtracting the mean Ct of the miRNA targets from the mean Ct value of the two housekeeping genes. The ∆∆Ct value was derived by subtracting the ∆Ct of the unknown sample from the ∆Ct of the reference sample (mean ∆Ct of R308 or RG). Finally, the fold change (2−∆∆Ct) was obtained.

The results are presented as the mean ± SD. GraphPad Prism 9 (Version 9.4.1 (458), GraphPad Software Inc., La Jolla, CA, USA) was used to plot the results. Data were analyzed using Student’s *t*-test. *p* < 0.05 was considered statistically significant, and *p*  <  0.01 was considered highly statistically significant (* *p* < 0.05; ** *p* < 0.01).

## 5. Conclusions

The miRNA, miR-126, has proven to be a promising biomarker in Ranger Gold lung tissue with statistically significant expression (*p* < 0.01). Studies in the literature have suggested that miR-126 might play a particularly important role in the lung, as it is normally highly expressed in lung tissue. Ranger Gold hybrids show significantly higher expression compared to R308 broilers, suggesting that the slow-growing hybrid has greater respiratory capacity and, consequently, a higher oxidative metabolism.

During the butchering process, the R308 broilers showed some anomalies, including airsacculitis, hepatic steatosis, and an enlarged spleen. By comparing healthy chickens with those having these issues, the levels of reference miRNA expression were evaluated to determine if these conditions influenced them. Chickens with airsacculitis and hepatic steatosis showed an up-regulation of miR-21 and miR-126 in the breast. These overexpressed miRNAs, compared to normal levels, are likely disease biomarkers, given their abnormal presence in the skeletal muscle of Ross 308, possibly due to the rapid growth of the animal and also the presence of a bacterial or viral infection.

This preliminary study was conducted on a limited sample of animals to avoid unnecessary slaughtering, in agreement with the poultry slaughtering and processing company. However, despite the limited extent of the sample set, we observed a trend in the data that was the result of prominent differences observed among sample sets. These findings are quite promising, and they can suggest the use of specific miRNAs as biomarkers for the identification of diseases in poultry breeds that might negatively influence meat quality and animal wellbeing. Future studies will focus on increasing either the number of animals/breeds and combination of animal status (e.g., healthy vs. diseased; types of disease; degree of disease condition) to ensure more robust and reliable results. Furthermore, future studies will also focus on other bioanalytical techniques more and more applied to food analysis (i.e., liquid chromatography-mass spectrometry), that could provide for a rapid, direct identification and determination of herein investigated biomarkers. This analytical technique and its use in omic approaches has the potential to develop a targeted quantitative method for screening specific miRNAs for different animal tissues, thus enabling an efficient and effective screening platform for meat quality and traceability.

## Figures and Tables

**Figure 1 molecules-29-00748-f001:**
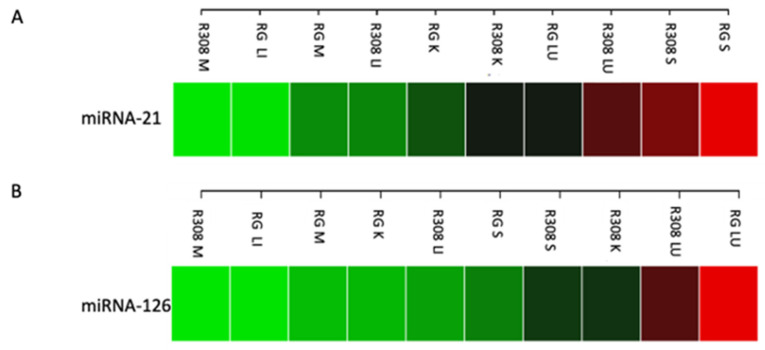
Heatmap showing the expression level of miR-21 (**A**) and miR-126 (**B**) in Ross 308 (R308) and Ranger Gold (RG). M, muscle (breast); LI, liver; K, kidney; LU, lung; S, spleen. Green, low expression—Red, high expression.

**Figure 2 molecules-29-00748-f002:**
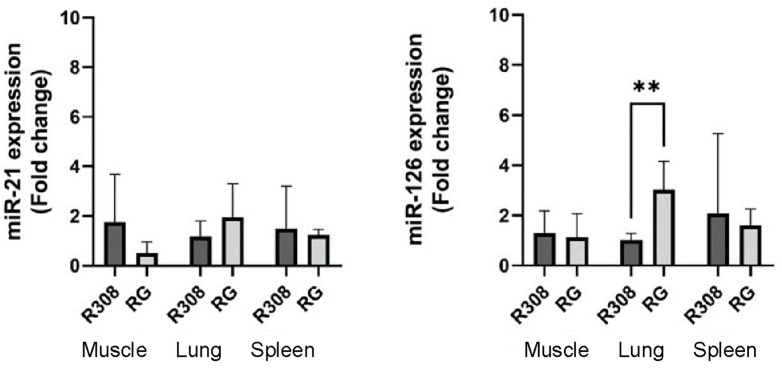
Expression levels of miR-21 and miR-126 in the two broilers (R308 and RG) in the three tissues: muscle (breast), lung, and spleen; the experiments were conducted with six conventional R308 chickens and five slow-growing RG chickens. Data are presented as means ± SD. ** symbol means statistically significant difference between bars (*p* < 0.01).

**Figure 3 molecules-29-00748-f003:**
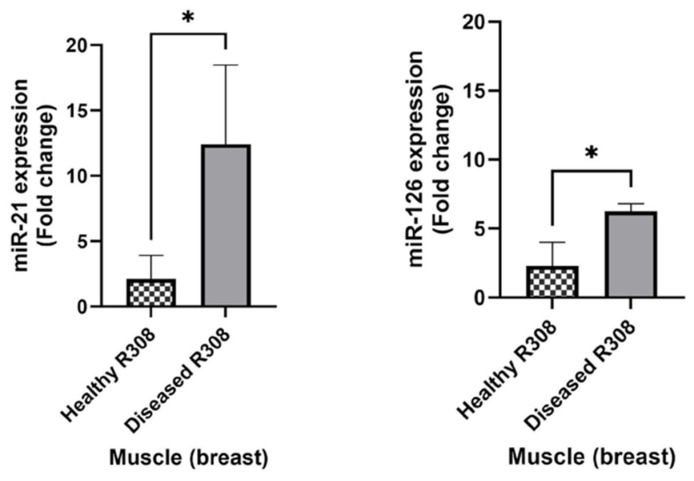
Mean values of relative expression levels in R308 healthy animals (*n* = 4) vs. animals presenting airsacculitis and hepatic steatosis (diseased, *n* = 2). * symbol means statistically significant difference between bars (*p* < 0.05).

**Table 1 molecules-29-00748-t001:** The table presents a detailed list of the analyzed animals, outlining their individual characteristics such as gender, breed, and other relevant attributes, along with corresponding extractions of small RNAs. * sampling date 4 June 2021; ** sampling date 18 January 2023.

ID	Breed	Average Weight	Sex	Tissue	Notes	RNA Extraction Date	RNA Concentration (ng/μL)	OD 260/280
FIRST ROUND SAMPLING *								
R308_1	Ross 308	unk	unk	Muscle (breast)	n/a	14 July 2021	230.04	2.13
R308_1	Ross 308	unk	unk	Liver	n/a	20 July 2021	141.74	2.14
R308_1	Ross 308	unk	unk	Spleen	n/a	21 July 2021	516.37	2.04
R308_1	Ross 308	unk	unk	Kidney	n/a	22 July 2021	1303.69	2.14
R308_1	Ross 308	unk	unk	Lung	n/a	27 July 2021	1764.78	2.12
RG_1	Ranger Gold	unk	unk	Muscle (breast)	n/a	16 July 2021	372.76	2.16
RG_1	Ranger Gold	unk	unk	Liver	n/a	20 July 2021	1573.03	2.17
RG_1	Ranger Gold	unk	unk	Spleen	n/a	21 July 2021	825.62	2.16
RG_1	Ranger Gold	unk	unk	Kidney	n/a	22 July 2021	962.51	2.14
RG_1	Ranger Gold	unk	unk	Lung	n/a	22 July 2021	567.03	2.09
SECOND ROUND SAMPLING **								
R308_1-B2	Ross 308	3.56	male	Muscle (breast)	hepatic steatosis	26 April 2023	220.77	2
R308_1-L2	Ross 308	3.56	male	Lung	hepatic steatosis	26 April 2023	198.83	2.07
R308_1-S2	Ross 308	3.56	male	Spleen	hepatic steatosis	29 April 2023	188.17	2
R308_2-B1	Ross 308	3.56	female	Muscle (breast)	n/a	20 March 2023	222.33	2.11
R308_2-L1	Ross 308	3.56	female	Lung	n/a	20 March 2023	166.05	2.01
R308_2-S1	Ross 308	3.56	female	Spleen	n/a	20 March 2023	234.68	2.05
R308_3-B1	Ross 308	3.56	male	Muscle (breast)	enlarged spleen	20 March 2023	122.03	2.26
R308_3-L1	Ross 308	3.56	male	Lung	enlarged spleen	20 March 2023	228.21	2.01
R308_3-S2	Ross 308	3.56	male	Spleen	enlarged spleen	29 March 2023	306.64	2
R308_4-B1	Ross 308	3.56	male	Muscle (breast)	airsacculitis	21 March 2023	161.47	2.12
R308_4-L1	Ross 308	3.56	male	Lung	airsacculitis	20 March 2023	201.68	2.04
R308_4-S1	Ross 308	3.56	male	Spleen	airsacculitis	21 March 2023	202.3	2.07
R308_5-B1	Ross 308	3.56	male	Muscle (breast)	airsacculitis	21 March 2023	235.38	2.11
R308_5-L1	Ross 308	3.56	male	Lung	airsacculitis	21 March 2023	145.54	2.05
R308_5-S1	Ross 308	3.56	male	Spleen	airsacculitis	21 March 2023	236.91	2.03
R308_6-B1	Ross 308	3.56	female	Muscle (breast)	n/a	21 March 2023	138.56	2.13
R308_6-L1	Ross 308	3.56	female	Lung	n/a	21 March 2023	132.3	2.1
R308_6-S1	Ross 308	3.56	female	Spleen	n/a	21 March 2023	253.44	2.04
RG_1-B1	Ranger Gold	3.14	female	Muscle (breast)	n/a	22 March 2023	175.81	2.08
RG_1-L1	Ranger Gold	3.14	female	Lung	n/a	21 March 2023	152.04	2
RG_1-S1	Ranger Gold	3.14	female	Spleen	n/a	21 March 2023	302.63	2.04
RG_2-B1	Ranger Gold	3.14	female	Muscle (breast)	n/a	22 March 2023	165.86	2.09
RG_2-L1	Ranger Gold	3.14	female	Lung	n/a	22 March 2023	159.73	2.03
RG_2-S1	Ranger Gold	3.14	female	Spleen	n/a	22 March 2023	182.07	2.09
RG_3-B1	Ranger Gold	3.14	female	Muscle (breast)	n/a	23 March 2023	151.64	2.22
RG_3-L1	Ranger Gold	3.14	female	Lung	n/a	22 March 2023	166.03	2.04
RG_3-S1	Ranger Gold	3.14	female	Spleen	n/a	23 March 2023	212.33	2.01
RG_4-B1	Ranger Gold	3.14	female	Muscle (breast)	n/a	23 March 2023	214.42	2.13
RG_4-L1	Ranger Gold	3.14	female	Lung	n/a	23 March 2023	233.54	2
RG_4-S1	Ranger Gold	3.14	female	Spleen	n/a	23 March 2023	154.55	2.04
RG_5-B1	Ranger Gold	3.14	female	Muscle (breast)	n/a	24 March 2023	146.81	2.17
RG_5-L1	Ranger Gold	3.14	female	Lung	n/a	23 March 2023	177.1	2.05
RG_5-S1	Ranger Gold	3.14	female	Spleen	n/a	24 March 2023	178.55	2.04

## Data Availability

The raw data supporting the conclusions of this article will be made available by the authors on request.

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
