# Peer review of "miRNAs as Biomolecular Markers for Food Safety, Quality, and Traceability in Poultry Meat—A Preliminary Study"

_molecules, 2024, doi:10.3390/molecules29040748_

Round 1

Reviewer 1 Report

Comments and Suggestions for Authors

In order to identify biomarkers for the traceability and quality of poultry meat, the expression and abundance of two candidate chicken miR-21 and miR-126 have been analyzed in this study. Results indicate that miR-126 could be a promising biomarker for the lung tissue in the RG breed, and a potential applicability for tracing hybrids. This is a quite interesting work, and the manuscript is well-organized. All the data is sufficient, and the figure/table is informative. I think it can be published with only some minor revision, such as add some information about 1) the research progress with the biomarkers in the Introduction, 2) the comparison with other studies in the Discussion.

Author Response

REVIEWER 1: I think it can be published with only some minor revision, such as add some information about 1) the research progress with the biomarkers in the Introduction, 2) the comparison with other studies in the Discussion.

We thank the reviewer for her/his suggestions and we accept them. The following changes has been applied to the manuscript text as requested by reviewer 1:

1) Introduction section was extensively revised and a brief paragraph was added (lines 71-116) with the aim of focusing more on investigation of miRNAs as biomarkers in food research. 8 new references were added to this section (16-23).

2) Regarding this second comment, we can point out that a relevant number of literature works has been already cited in the Discussion section for covering all the peculiar topics of our findings, such as: comparison with other research articles about the use of housekeeping genes employed for the result normalization (refs. 37-41); role and regulation activities of miR-126 (refs- 33, 45-49) and miR-21 (refs. 56-59), and their connection with chicken health status (refs. 32, 50-51 for miR-126; refs. 52-55 for miR-21). However, we included three more bibliographic references in the discussion section (refs. 42-44) as response to another reviewer’s comment (lines 225-226) about no sex-specific expression of miR-126 and miR-21 in the tissues object of this study. We hope this could be sufficient for addressing the reviewer comment.

Reviewer 2 Report

Comments and Suggestions for Authors

In the manuscript titled “miRNAs as Biomolecular Markers for Food Safety, Quality, and Traceability in Poultry Meat – a Preliminary Study,” the authors identified the expression and abundance of two candidate miRNAs – miR-21 and miR-126 to be used as potential biomarkers for traceability and quality of poultry meat. The expression levels of these miRNAs were measured on spleen, lung, and muscle (breast) tissues collected from two breeds – Ross308 and Ranger-Gold. The manuscript and research question addressed here are interesting; however, I have some major concerns:

1.     The authors initially gathered five tissues from two samples during round 1 (Muscle, liver, kidney, lung, and spleen) and three tissues from round 2 (Lung, spleen, and muscle). Round 1 involved two samples, each from a different breed. However, it's crucial to note that for any statistically significant findings, a minimum of three samples is generally required for conclusive analysis. To address the use of only one sample for generating preliminary data, the authors should provide a compelling justification for this approach and explain how it informed the subsequent tissue collection strategy.

2.     Numerous RNA-Seq studies indicate sexually dimorphic gene expression in chickens across various tissues. The authors need to address the potential effects of sex, particularly considering the unbalanced male-female distribution in the R308 group (4 males and 2 females) compared to the RG group (5 females). The observed higher expression of miR-126 in RG lungs compared to R308 broilers might be influenced by sex effects. To ensure the robustness of their conclusions, the authors are advised to discuss and account for these sex-related variations in their study.

3.     It is strongly recommended that the authors balance the sex distribution by including an equal number of males and females in both the RG and R308 groups for both the miRNAs to enhance the relevance of their conclusions and ensure the study's validity.

4.     Consider moving Table 1 to the Supplementary section. Additionally, correct the typo "Muslce" to "muscle."

5.     In Line 228, there is a typo. Replace the "e" in "miR-21 e miR-126" with "and" for clarity and accuracy.

6.     In Line 249, the authors reported anomalies in two chickens from the R308 group, including airsacculitis, hepatic steatosis, and an enlarged spleen. However, Table 1 indicates four R308 chickens as unhealthy and two as healthy. A thorough examination of the dataset is needed to reconcile these discrepancies and ensure the accuracy of the reported findings.

7.     Considering the small sample size in one group for Figure 3, performing a student t-test may not be ideal. The authors should justify their choice of statistical test and consider alternative approaches or analyses suitable for a limited sample size. It's essential to address the potential limitations and uncertainties associated with statistical analyses based on a small number of samples to ensure the robustness of the study's conclusions.

Author Response

REVIEWER 2:

1. The authors initially gathered five tissues from two samples during round 1 (Muscle, liver, kidney, lung, and spleen) and three tissues from round 2 (Lung, spleen, and muscle). Round 1 involved two samples, each from a different breed. However, it's crucial to note that for any statistically significant findings, a minimum of three samples is generally required for conclusive analysis. To address the use of only one sample for generating preliminary data, the authors should provide a compelling justification for this approach and explain how it informed the subsequent tissue collection strategy.

This is a valid question, and we apologize we haven’t explained this better in the text. In the initial project hypothesis of work the tissue to be tested for obvious commercial value was the breast muscle. We sampled two breeds (i.e., RG and R308) and a set of tissues (e.g., muscle, liver, kidney, lung, and spleen) to look at general expression trends. First, we did the RNA extraction and we found that the RNA quality in all tissues was very good and comparable across tissues. The following step was to detect the expression of the miRNAs and to optimize the qPCR protocol. The finding of higher expression in the lung and the spleen for miRNA-126 and miRNA-21 respectively was very interesting and worth further exploration. That is why in the second sampling step we focused not only on the muscle, but we also tested lung and spleen tissues. The sentence at line 157 has been modified in the text (under section 2.2 “Analysis and quantification of the presence of two specific miRNAs”) as in the following: “Remarkably, in both breeds, the miRNA-21 was strongly expressed in spleen (Fig. 1A), while miRNA-126 was strongly expressed in the lung (Fig. 1B), with higher values in RG compared to R308. For this reason, in the second sampling procedure, the lung and the spleen tissues were collected and further analyzed together with the muscle tissue”.

2. Numerous RNA-Seq studies indicate sexually dimorphic gene expression in chickens across various tissues. The authors need to address the potential effects of sex, particularly considering the unbalanced male-female distribution in the R308 group (4 males and 2 females) compared to the RG group (5 females). The observed higher expression of miR-126 in RG lungs compared to R308 broilers might be influenced by sex effects. To ensure the robustness of their conclusions, the authors are advised to discuss and account for these sex-related variations in their study.

We thank the reviewer for the legitimate comment, and we agree on the limitations of sample size and gender balance. Unfortunately, we were not able to choose the gender in advance nor the number of the animals that were sacrificed (line 367); also, we conducted a literature search when choosing the miRNAs to test in this study, to make sure that gender would not be an issue. miRNAs can in fact be differentially expressed in animals according to the gender, and that is mostly true in gonadic tissues. However, miRNA-21 and miRNA-126 do not seem to be among the most gender-specific miRNAs in chicken (e.g., Zou X, Wang J, Qu H, Lv XH, Shu DM, Wang Y, Ji J, He YH, Luo CL, Liu DW. Comprehensive analysis of miRNAs, lncRNAs, and mRNAs reveals potential players of sexually dimorphic and left-right asymmetry in chicken gonad during gonadal differentiation. Poult Sci. 2020 May;99(5):2696-2707. doi: 10.1016/j.psj.2019.10.019). 

Some other tissues influenced by hormonal signaling are affected by miRNA differential expression but to our knowledge there is no sex-specific link of miRNA-21 and miRNA126 with gender in mouse, human and chicken brain, heart, liver and gonads  (Warnefors M, Mössinger K, Halbert J, Studer T, VandeBerg JL, Lindgren I, Fallahshahroudi A, Jensen P, Kaessmann H. Sex-biased microRNA expression in mammals and birds reveals underlying regulatory mechanisms and a role in dosage compensation. Genome Res. 2017 Dec;27(12):1961-1973. doi: 10.1101/gr.225391.117) nor in diseases (Sharma S, Eghbali M. Influence of sex differences on microRNA gene regulation in disease. Biol Sex Differ. 2014 Feb 1;5(1):3. doi: 10.1186/2042-6410-5-3).

We have added the following text in the revised manuscript to explain the limitation of our dataset and the significance of the results we have obtained (line 225): “To our knowledge the miRNA analyzed in this study do not show sex-specific expression in the tissues tested in both physiological and diseased state”. The above mentioned articles were added in the bibliography as references 42-44.

3. It is strongly recommended that the authors balance the sex distribution by including an equal number of males and females in both the RG and R308 groups for both the miRNAs to enhance the relevance of their conclusions and ensure the study's validity.

We thank the reviewer for this remark and we would like to stress the fact that we agree with her/him. As explained in the previous response, to our knowledge there is no sex-specific link of the miRNA tested with gender nor disease. As for sample size, that variable has not much to do with observed effects (whether or not they are true), but only with failing to observe an effect (whether or not one exists). This can be fixed in different ways: (i) increase sample size or (ii) set a different statistical significance level. We were not able to increase the sample size but we tried to analyze data going below routinely used alpha level value (below 0.05) to exclude the probability of a false positive result. However, we agree that the number of samples could be a limit in our study which has its most value in the development and optimization of the technique and the role of the potential miRNA selected.

4. Consider moving Table 1 to the Supplementary section. Additionally, correct the typo "Muslce" to "muscle."

The typo In Table 1 (materials and methods section) has been corrected.

We think that Table 1 contains lots of useful information needed by the reader in the main text. For this reason we did not move Table 1 to the Supplementary Material.

5. In Line 228, there is a typo. Replace the "e" in "miR-21 e miR-126" with "and" for clarity and accuracy.

We corrected the typo in the manuscript text as evidenced by this reviewer.

6. In Line 249, the authors reported anomalies in two chickens from the R308 group, including airsacculitis, hepatic steatosis, and an enlarged spleen. However, Table 1 indicates four R308 chickens as unhealthy and two as healthy. A thorough examination of the dataset is needed to reconcile these discrepancies and ensure the accuracy of the reported findings.

We agree with the Reviewer for this accurate observation. During the slaughtering process, it was noted that four R308 chickens exhibited anomalies, including airsacculitis, hepatic steatosis, and an enlarged spleen. Among these four chickens, two of them, despite one having airsacculitis and the other an enlarged spleen, did not show significant differences in relative expression compared to healthy R308 chickens (the results were comparable) and were therefore considered healthy. On the other hand, the remaining two chickens, one with airsacculitis (accompanied by diarrhea) and the other with hepatic steatosis, showed a much higher relative expression and were consequently classified as diseased. We have also addressed this reviewer’s observation in the section "2.4. Expression of miRNAs in healthy and diseased R308 broilers" by revising the manuscript text from lines 191 to 197.

7. Considering the small sample size in one group for Figure 3, performing a student t-test may not be ideal. The authors should justify their choice of statistical test and consider alternative approaches or analyzes suitable for a limited sample size. It's essential to address the potential limitations and uncertainties associated with statistical analyses based on a small number of samples to ensure the robustness of the study's conclusions. 

We thank the Reviewer for her/his careful observation. We are aware of the limitations of the student t-test and acknowledge that with a limited number of samples there is no test that can guarantee definitive statistical relevance and significance. We have conducted additional analyses using both parametric and non-parametric methods but we considered the t-test to be the most appropriate approach for this data.  In fact, despite the limitations, a clear trend between sample sets due to prominent differences could be observed.  

It is also crucial to point out that the initial goal of the article was not studying disease in chickens. However, since we identified remarkable differences in the relative expressions of the miRNA analyzed in healthy vs diseased chickens, we deemed it crucial to include such observations. The most important finding of this work is the validity of the method optimized and employed and the interest of the miRNA selected as biomarkers in poultry farming. This observation propels us in the right direction for further research on the potential utility of these results as a foundation for biomarker development. Anyway, we changed the sentence in the conclusion in line 367 to include the following considerations: “This preliminary study was conducted on a limited sample of animals to avoid unnecessary slaughtering, in agreement with the poultry slaughtering and processing company. However, despite the limited extent of the sample set, we observed a trend in the data, that was the result of prominent differences observed among sample sets”.

Round 2

Reviewer 2 Report

Comments and Suggestions for Authors

Thank you authors for addressing all the comments.